# A Transcriptomic Meta-Analysis Shows Lipid Metabolism Dysregulation as an Early Pathological Mechanism in the Spinal Cord of SOD1 Mice

**DOI:** 10.3390/ijms22179553

**Published:** 2021-09-02

**Authors:** Luis C. Fernández-Beltrán, Juan Miguel Godoy-Corchuelo, Maria Losa-Fontangordo, Debbie Williams, Jorge Matias-Guiu, Silvia Corrochano

**Affiliations:** 1Neurological Disorders Group, Hospital Clínico San Carlos, IdISSC, 28040 Madrid, Spain; luisferbel18@hotmail.com (L.C.F.-B.); juanmiguelgodoycorchuelo@gmail.com (J.M.G.-C.); marlosa@ucm.es (M.L.-F.); jorge.matiasguiu@salud.madrid.org (J.M.-G.); 2Mammalian Genetics Unit, MRC Harwell Institute, Oxfordshire OX11 0RD, UK; d.williams@har.mrc.ac.uk

**Keywords:** ALS, *SOD1*, RNA-seq, lipid metabolism, meta-analysis, cholesterol

## Abstract

Amyotrophic lateral sclerosis (ALS) is a multifactorial and complex fatal degenerative disorder. A number of pathological mechanisms that lead to motor neuron death have been identified, although there are many unknowns in the disease aetiology of ALS. Alterations in lipid metabolism are well documented in the progression of ALS, both at the systemic level and in the spinal cord of mouse models and ALS patients. The origin of these lipid alterations remains unclear. This study aims to identify early lipid metabolic pathways altered before systemic metabolic symptoms in the spinal cord of mouse models of ALS. To do this, we performed a transcriptomic analysis of the spinal cord of *SOD1^G93A^* mice at an early disease stage, followed by a robust transcriptomic meta-analysis using publicly available RNA-seq data from the spinal cord of SOD1 mice at early and late symptomatic disease stages. The meta-analyses identified few lipid metabolic pathways dysregulated early that were exacerbated at symptomatic stages; mainly cholesterol biosynthesis, ceramide catabolism, and eicosanoid synthesis pathways. We present an insight into the pathological mechanisms in ALS, confirming that lipid metabolic alterations are transcriptionally dysregulated and are central to ALS aetiology, opening new options for the treatment of these devastating conditions.

## 1. Introduction

Amyotrophic lateral sclerosis (ALS) is a fatal neurodegenerative disorder affecting primarily the degeneration of upper and lower motor neurons leading to muscle atrophy, paralysis, and death [1]. The great majority of cases are sporadic (around 90–95%) with the remaining 5–10% familial in nature (fALS) caused by different mutations. The most frequent mutation is a toxic hexanucleotide repeat expansion in the *C9ORF72* gene, accounted by up to 35% of familial cases, followed by mutations in the Cu/Zn-superoxide dismutase 1 (*SOD1*) gene (15–30% of fALS and up to 2% of the total of ALS cases) [2,3], although these numbers might be underestimated due to the current limitations of genetic testing in sporadic cases. There are over 30 genes identified that are causative or modifiers of ALS. These genes have been crucial for understanding the aetiology of disease, providing evidence of the complexity and multifactorial nature of this complex disorder [4]. Many processes are implicated in the pathomechanism of ALS, including neuroinflammation, RNA processing and metabolism, axonal transport, vesicle trafficking, proteostasis, oxidative stress and mitochondrial dysregulation [5]. There is also increasing evidence that lipid homeostasis alterations are important in the pathological disease process in ALS.

Lipids are essential for the structure and function of neuronal tissue. The brain has very high lipid content, second only to adipose tissue, comprising of 50% of the brain dry-weight [6]. It is therefore not surprising that lipid metabolic alterations could be at the core of pathological processes in neuronal tissue. Alterations in lipid metabolism manifest in ALS, especially once the disease progresses to late stages. Thus, until recently, lipid metabolism disturbances were perceived as a consequence of disease progression. The most apparent observation has been the associated weight loss in ALS patients and mouse models, correlating with disease progression [7,8,9]. On the other hand, hyperlipidaemia is associated with better prognosis [10]. Thus, increasing the fat content in diets has been used in mouse models of ALS, reporting an extension in survival [11]. Accordingly, this has urged clinical trials aiming to restore the energy balance with diets rich in fats [12,13], relatively successful, although more extensive studies and larger cohorts are needed to draw more definitive conclusions. Lipid alterations have been observed in blood and cerebrospinal fluid (CSF) of ALS patients and in a symptomatic mouse model of ALS overexpressing mutant human Cu/Zn-superoxide dismutase gene (*SOD1^G93A^*) [14], correlating with disease prognosis [15]. The neuronal tissues also show alterations in lipid metabolism, especially in the spinal cord tissues of *SOD1^G93A^* rats [16] and mice, as well as in post mortem spinal cord tissue of ALS patients [17].

The question whether lipid homeostasis alterations are a consequence of disease or have a causal role in the neurodegenerative process still remains unresolved. However, there is evidence supporting that, in fact, lipid homeostasis could be an early pathological event in the aetiology of ALS. For instance, a proteomic analysis of the spinal cord of *SOD1^G93A^* mice at pre-symptomatic disease stage identified early alterations in glycolysis, β-oxidation, and mitochondrial metabolism, before disease onset [18]. In relation to human data, we can observe an association of risk susceptible genes involved in lipid metabolic pathways, such as CYP27A1 in sporadic ALS, suggesting that lipid dysregulation might be causally linked to ALS [19].

The *SOD1^G93A^* mouse is the most widely used animal model of ALS. It carries multiple copies of a human *SOD1* transgene with the ALS mutation G93A. The disease progression is well characterized in this mouse model. The onset of systemic body changes, in particular weight loss, starts at around 100 days of age, although the motor neuron degeneration in the spinal cord starts as early as two months of age (around 60 days). The disease progresses with weight loss and degeneration of the spinal cord, ending in the paralysis of the legs after 120 days, and death around 150 days of age. Thus, it is a useful model of lower motor neuron disease and also a fast progressive model to use in an experimental setting [20].

The use of RNA-sequencing has brought an enormous value to science as it allows for an unbiased analysis of the gene expression profiles in different tissues, cells, and contexts, helping to unravel important biological processes operating under different conditions. One of the main limitations of this technique is the variability in results among the different laboratories, which could be the result of the particular sequencing technique of choice, the RNA extraction method, the number of samples used, and other limitations coming from the different nature of the samples. Therefore, only the most highly altered genes are consistently reported across different studies. To overcome this problem, the combination of several databases, e.g., a meta-analysis using data generated by different laboratories, gives more robust results.

Herein, we look for early transcriptionally regulated lipid metabolic pathways that might bring new insights into the causal or sequential order of lipid changes previously reported in the degenerating spinal cords of mice and ALS patients. We firstly conducted RNA-sequencing followed by an unbiased transcriptomic analysis of the spinal cord of *SOD1^G93A^* females at 90 days (P90), before the onset of the body weight loss (early symptomatic disease stage) and then focused on the identification of the altered genes involved in the regulation of lipid metabolism. In order to strengthen our data, we conducted two meta-analyses using other publicly available transcriptomic databases of spinal cord SOD1 mice at early symptomatic, and also, late symptomatic (around P120) stages of disease. The meta-analyses evidenced that altered regulation of several lipid metabolic pathways are early events in the degeneration of the spinal cord of SOD1 mice that exacerbate as disease progresses. We further identified several lipid related genes and pathways, e.g., the *Ch25h* gene that shows transcriptional dysregulation from very early disease onset (P60), which might be important therapeutic targets.

## 2. Results

### 2.1. Analysis of Transcriptional Profile of the Spinal Cord from Early Symptomatic Disease Stage SOD1^G93A^ Mice

To identify early changes in lipid metabolic regulation in the spinal cord of *SOD1^G93A^* mice, we performed RNA-sequencing analysis of female *SOD1^G93A^* and wild type littermate mice (*n* = 5) at early symptomatic disease stage (90 days of age, P90), which was before the onset of body weight loss. We found a total of 1173 genes differentially expressed in the spinal cord of *SOD1^G93A^* mice compared to wild type controls, with log2-fold change values between −1.5 and +1.5 (Figure 1A, and Appendix A). A total of 639 genes were upregulated and 535 were downregulated. Among the top significant hits, we identified some previously confirmed genes related to *SOD1* pathology, and some of those genes (*Trem2*, *Nefh*, *C1q*, *Ccl6*, and *Mmp9*) are highlighted in the volcano plot (Figure 1A). In addition, a principal component analysis (PCA) showed that 39% of the variability among samples was due to genotype. The moderate percentage of variability is consistent with the early symptomatic stage of the samples (Figure 1B). The heatmap of identified differentially expressed genes (DEG) showed consistent patterns of up- or downregulated genes across the samples grouped by genotype (Figure 1C).

Next, we performed an unbiased functional enrichment analysis (Gene Ontology—GO) of the differentially expressed genes (DEGs). Analysis of upregulated genes showed that the most significantly upregulated biological processes were related to immune system functions (Table 1). This is consistent with previous studies and confirms that neuroinflammation is an early event in the pathology of SOD1 transgenic mice. The most significantly downregulated biological processes were related to neuronal function including axon transport and synapses (Table 1). We looked for significant GO terms in relation to lipid metabolism and found the following: “GO:0055094 response to lipoprotein particle”, “GO:0036314 response to sterol”, “GO:0019216 regulation of lipid metabolic process”, and “GO:0046486 glycerolipid metabolic process” (Table 2). Interestingly, all genes relating to lipid metabolic GO terms were upregulated.

### 2.2. “Lipid Metabolism” Is a Biological Process That Is Transcriptionally Dysregulated in the Spinal Cord of SOD1^G93A^ Mice at P90

The gene ontology analysis found few processes and genes related to lipid metabolism, although there are other genes involved in the regulation of the lipid metabolism that were not detected by the gene ontology analysis. This is a limitation to the identification of biological processes, pathways, and genes with a small but relevant significant meaning in a particular process. There is a minimum number of significantly dysregulated genes in order to make a significant GO term. Therefore, many lipid-related genes and processes might be missed by this type of analysis. In this sense, it is relatively easy to identify robust changes at symptomatic disease stages, but it is normally more difficult to identify very early changes in early symptomatic or pre-symptomatic stages. To this end, we used a targeted approach to find the total number of lipid related genes among the DEGs in our study.

For the targeted identification of lipid related genes, we first looked for all the genes that are annotated in the genome within the category of lipid metabolism. We merged the lists of genes annotated in the GO term “lipid metabolism process” (GO:0006629) and “lipid transport” (GO:0006869), resulting in a total of 1587 genes annotated in the genome as lipid related genes (Appendix A). Using this target list of 1587 lipid genes, we then compared it to the 1173 DEGs from our study. We found that a total of 127 genes (74 upregulated and 53 downregulated) were lipid metabolism genes altered at P90 in the spinal cord of *SOD1^G93A^* mice (Figure 2).

### 2.3. Meta-Analysis of RNA-seq Datasets from Spinal Cord of SOD1 Mice at Early and Late Symptomatic Disease Stages

Changes in lipid metabolic pathways have been more consistently identified by different groups in late symptomatic stages of disease, in the spinal cords of both mice and ALS patients [16,21]. However, due to the difficulty of determining very early changes in disease, the variability of the findings in early symptomatic stages of disease in the spinal cord among groups is high. The lipid metabolic changes identified above are relatively mild and, knowing that there is high variability among different RNA-seq studies and among labs, we decided to corroborate and strengthen our data with other RNA-seq studies performed on spinal cord samples of SOD1 mice at the same early symptomatic disease stage (P90), and performed a meta-analysis of the combined studies, including our own. We also decided to compare the findings between the early (P90) and late symptomatic (from 120 days, P120) disease stages, in order to determine the transcriptional regulation involved in the well-described lipid alterations at late disease stages.

First, we needed to find all the publicly available transcriptomic databases that fulfil the criteria to be included in the early symptomatic meta-analysis. Following the exact criteria: early symptomatic stage (around P90) AND spinal cord AND SOD1 mutant mice AND bulk RNA-seq, we could only find two RNA-seq datasets publicly available that could be used in combination to our own study (study 1) for the meta-analysis (study 2 and study 3, Table 3). For the comparative meta-analysis of late symptomatic disease stage, we established the following criteria: late symptomatic, AND spinal cord, AND SOD1 mice, AND bulk RNA-seq. In the search of publicly available resources, we found three RNA-seq studies (studies 4, 5 and 6) (Table 3).

The raw data from all these studies were downloaded and re-analysed following the same pipeline that we used for our RNA-seq study. The analysis identified a total of 721 and 1273 DEGs in the spinal cords of the SOD1 animals in the study 2 and 3, respectively (Figure 3A–C), which is within a similar range to our RNA-seq study 1 (1173 DEGs). The individual analysis of the raw data of the RNA-seq from the three late symptomatic studies found many more DEGs than in the early symptomatic stage, as expected. In particular, a total of 4623 DEGs were identified in study 4, a total of 9543 DEGs in study 5, and only 1272 DEGs in study 6 (Appendix A). These results manifest the vast heterogeneity found among studies, and support the need to conduct more robust combined meta-analyses when possible.

Next, we performed two separated meta-analyses; one using the data from the DESseq2 analysis from each of the three early symptomatic studies, and another with the data from the DESseq2 analysis from each of the three late symptomatic studies. The early symptomatic meta-analysis (Early MA) identified 1520 DEGs (Figure 3D and Appendix A), which is more than that found by any of the individual studies. The fact that the Early MA found more DEGs than any of the individual studies is not surprising, as the use of different studies increases the sample size and the statistical power, thus resulting in the identification of genes that might not be found in any of the studies separately. From those 1520 DEGs of the Early MA, 839 DEGs were also found in our original study 1, and 75 of the genes found in the meta-analysis were not identified in any of the three studies. The late symptomatic meta-analysis (Late MA) found 7416 DEGs (Appendix A).

### 2.4. Meta-Analysis of RNA-seq Datasets Evidences Alterations in Lipid Metabolic Processes in the Spinal Cord of SOD1 Mice

In the search for specific and robust changes in lipid related genes in the spinal cord at early and late stages of disease, we conducted a targeted approach. We looked for all the DEGs in the two meta-analyses that were lipid metabolic related genes, by comparing them to the list of 1587 lipid genes annotated in the genome by GO terms (Appendix A). These comparisons identified 166 lipid-related DEGs from the Early MA, as shown in the Venn diagrams (Figure 4, and Appendix A) and 690 lipid-related DEGs from the Late MA. Interestingly, our initial individual study 1 (from Figure 2) found 127 lipid related genes, and 93 of those are coincident with the 166 lipid related DEGs from the Early MA. Thus, using the meta-analysis, we are identifying more changes in the expression of lipid related genes at early stages of disease. Most of the Early MA lipid DEGs were present, and increasing, in the lipid DEGs from the Late MA (approx. 95%) supporting the validation of those findings.

### 2.5. Metabolic Pathways of Cholesterol, Phospholipids, Ceramides, and Icosanoids Are Transcriptionally Altered at Early Symptomatic Disease Stage in the Spinal Cord of SOD1 Mice

Next, we looked for the relation of those lipid related genes at P90, to try to identify the most relevant lipid metabolic pathways transcriptionally altered. Thus, the list of 166 Early MA lipid-related DEGs were analysed by STRING, a protein–protein interaction network. In the analysis, a few clusters or nodes were emergent (Figure 5). We identified those clusters with a functional enrichment analysis. The most relevant GO terms provided by the analysis (avoiding GO terms of general lipid processes) were coloured as follows: “GO:0008203 Cholesterol metabolism process” (red), “ GO:0006665 Sphingolipid metabolic process” (blue), “GO:0006690 eicosanoid metabolic process” (yellow), “GO:0008654 Phospholipid biosynthetic process” (cyan), “GO:0046488 phosphatidylinositol metabolic process” (pink), and “GO:0070372 regulation of ERK1 and ERK2 cascade (green)”.

Analysis of the 690 Late MA DEGs identified many of the same lipid related pathways found in the Early MA data, confirming once again that certain pathways are initiated at early stages of disease. The Late MA analysis confirmed that important lipid metabolism pathways such as cholesterol metabolism, ceramide catabolism, eicosanoid synthesis and phospholipid metabolism were highly transcriptionally dysregulated (Appendix A), showing more genes altered on each of the nodes/pathways identified than at the early symptomatic disease stage.

### 2.6. The Transcriptional Dysregulation of Lipid Pathways in the Spinal Cord of SOD1 Mice Is Exacerbated at Late Disease Stage

As described above, at the late symptomatic disease stages we found a higher number of genes altered in each of the main lipid pathways identified when compared to the early symptomatic P90. We next examined whether the level of expression of the genes identified at P90 could be more severely affected at late symptomatic stage. Thus, we selected the group of genes identified in each of the altered lipid pathways from the Early MA and represented their gene expression levels with a comparative early symptomatic and late symptomatic heatmap.

The heatmap of genes related to sterol-cholesterol clearly manifests that, at late symptomatic disease stage, there is more downregulation of the biosynthesis process, whereas there is a higher upregulation of the transport of cholesterol and other downstream processes (Figure 6A). The representation of these genes in the pathways points towards a general transcriptional downregulation of the enzymes involved in the process of cholesterol biosynthesis. This pathway is more transcriptionally repressed at late symptomatic disease stage (Figure 6B).

Similarly, the transcriptional changes in the glycosphingolipids and ceramides pathways (Figure 7A) were also exacerbated at late symptomatic disease stage. There are four pathways involved in the metabolism of ceramides: de novo biosynthesis from serine and palmitoyl CoA; sphingomyelin pathway; glycolipids pathway; and sphingosine-1P pathway. The representation of these genes in these pathways confirmed the upregulation of the catabolism of glycosphingolipids and glucosylceramides, suggesting that the production of ceramides by these pathways are a main feature (Figure 7B). Interestingly, the catabolic pathway of ceramides through the upregulation of the sphingosine-1P pathway, with the upregulation of *Asah1* and *Spgl1* genes, seems to be transcriptionally initiated from early disease stage. Of note, the de novo biosynthesis of ceramides is not upregulated, at least at early symptomatic disease stages in the spinal cord of SOD1 mice, and neither is the sphingomyelin pathway.

Finally, the eicosanoid pathways were also even more transcriptionally dysregulated at late symptomatic compared to early symptomatic disease stages, with a clear upregulation of the pathways towards an increase in biosynthesis of inflammation metabolites (Appendix A).

### 2.7. The Ch25h Gene Is an Early Marker of Lipid Alteratios in the Spinal Cord of SOD1 Mice

We performed validations of the gene expression level by quantitative PCR with a different set of biological samples using the lumbar region of the spinal cord. We included an even earlier and a later time point in order to evaluate the level of changes associated to those different disease stages. Therefore, we extracted RNA from the lumbar region of the spinal cord of *SOD1^G93A^* females at different ages: at 60 days of age (P60), at early symptomatic (90 days of age, P90) and at late symptomatic disease stage (at 120 days of age, P120). We confirmed the findings for the genes *Abca1* and *Ch25h* of the cholesterol pathways (Figure 8A), and the genes *Hexb* and *Sgpl1* of the ceramides pathway (Figure 8B). Most of the genes showed significant alteration from P90, validating the results from our RNA-seq study 1 and the results of the meta-analyses. Interestingly, the gene *Ch25h* shows significant upregulation in the spinal cord of *SOD1^G93A^* mice as early as 60 days of age, which could indicate that this gene might be one of the earliest initiators of the changes found in the cholesterol pathway (Figure 8A). The expression of this gene increases as the disease progresses, with the highest levels found at P120 in the *SOD1^G93A^* mice.

All these data confirmed that the alterations in lipid metabolism are strongly transcriptionally dysregulated in late symptomatic stages of disease, and evidenced that most of these alterations are initiated from early symptomatic stages.

## 3. Discussion

RNA sequencing techniques are useful for the determination of the transcriptome of particular tissues/cells at particular times and locations. Even though sequencing is becoming more affordable, the cost still limits the number of biological replicates and sequencing depth used by most researchers, compromising the detection power of differential gene expression between samples. The total number of samples is also influenced by the welfare needs of using the minimum number of animals that can yield statistical results. On top of this, the results might differ when the same experiments are conducted in different labs, mostly due to the variability in sequencing techniques, in the analysis, the exact starting material and especially in the way the samples are processed in the different laboratories. Therefore, similar transcriptomic analysis performed by different labs could yield different results. By performing meta-analysis of comparable RNA-seq data we are increasing the biological replicates and reducing the variability in the analysis, thus increasing the detection power [22]. This technique is particularly useful to study early stages of disease, where the differences in the expression of genes are normally very low, which increases the variability in the findings among different labs.

The main aim of this study was to identify the early and late transcriptional alterations in lipid metabolic pathways in the spinal cord of diseased SOD1 mice. Thus, we performed our own RNA-seq study from the spinal cord of *SOD1^G93A^* and wild type littermate females at an early symptomatic disease stage (P90). As expected from this early stage, most differential gene expressions were of low magnitude, with fold changes no bigger than +/− 1.5. We specifically looked for changes in lipid metabolic pathways, but we felt that the power of detection from one single experiment was limited. Thus, we decided to increase the detection power by gathering data from other early symptomatic spinal cord SOD1 RNA-seq studies, and performing a meta-analysis alongside our own study. At the same time, we gathered another three RNA-seq databases from late symptomatic disease stages of SOD1 spinal cord, and performed another transcriptomic meta-analysis, with the idea of finding a fraction of differentially expressed genes that might be presented in one disease stage but not necessarily the other, and also corroborate well-known lipid alterations previously reported at late disease stages.

Other processes have been implicated in the aetiology of the disease of different forms of ALS, leading to the multifactorial model proposed for ALS. The top significant pathways when performing an RNA-seq experiment related to ALS/neurodegeneration are normally associated to inflammation. Thus, although lipid metabolic pathways are also significant in many analyses, those are normally overlooked as they do not fall in the top pathways identified. We then followed this analysis with a targeted approach, to avoid missing any other important DEGs and pathways involved in lipid metabolism that might not be identified by GO terms. By using early symptomatic and late symptomatic datasets we could detect changes in lipid related gene expression profiles.

Alterations in cholesterol metabolism have been previously associated with ALS. Most studies have been performed at the systemic level, using serum and CSF, at early symptomatic and late symptomatic disease stages, and very few studies are conducted directly on the spinal cord and brain tissue. When using spinal cord and brain tissue from patients, the studies are limited to the end-stage of disease (i.e., post mortem tissues). The advantage of using mouse models of disease is that the study of neuronal tissue can be performed at any stage of disease. In serum and CSF, the total cholesterol levels are higher in ALS patients when compared to healthy controls, although some studies report lower blood levels of LDL cholesterol in patients [23,24] and mouse models [25]. It is not entirely clear whether levels of cholesterol in ALS are beneficial or detrimental. For instance, LDL cholesterol has been associated with increasing ALS risk by mendelian association studies [26], but at the same time, higher levels of LDL have been associated with a slower disease progression in ALS patients [10].

The source of higher cholesterol levels is not clear, especially at early disease stages. The dysregulation of lipids in ALS has been associated with the degenerative process. Once the neurons and neuronal tissue die, the cholesterol is released, which could explain the higher levels of cholesterol and cholesterol metabolites (in particular high levels of cholesterol esters and oxysterols) in the CSF and plasma [27,28] and spinal cord [29] in ALS patients and in mouse models at symptomatic disease stages [16,17,21]. On the contrary, what we have established in this study is that the control of cholesterol pathways are transcriptionally altered before the general degenerative process, which means that the altered levels of cholesterol are not solely a mere consequence of cell death. We have identified that cholesterol transport outside neuronal tissue is transcriptionally upregulated from early disease stages, which might explain early findings of cholesterol in the CSF, and not so evident in the plasma. Cholesterol cannot pass the blood–brain barrier [30,31,32], thus, the cholesterol used by neuronal tissue is mainly locally synthesized. In this work, we found that the local cholesterol biosynthesis is downregulated in the spinal cord from early disease stages. Taken together, the transcriptional profile results at early symptomatic spinal cord of SOD1 transgenic mice suggest an attempt to reduce the endogenous levels of cholesterol, by repressing the biosynthesis and upregulating the transport pathways. We also identified the main genes that are altered in those pathways at early stages, which might represent potential therapeutic targets. The rate-limiting enzyme in cholesterol biosynthesis HMG-CoA reductase, encoded by the *HMGCR* gene, has been found to be reduced in the spinal cord of ALS patients [29], and in this work, we have found that this gene is transcriptionally downregulated from early disease stages. Interestingly, statins, the cholesterol lowering drugs that inhibit HMG-CoA enzyme activity, have been controversially suggested to be detrimental for mice and ALS patients [33,34]. Another crucial gene that we have identified in our study is *CH25H*. Interestingly, this gene appears significantly upregulated from pre-symptomatic disease stage P60, representing one of the earliest genes altered from the cholesterol pathways. Hence, this gene could be an interesting target that deserves further attention. The *CH25H* gene encodes the cholesterol 25-hydroxylase enzyme that converts cholesterol into 25-hydroxycholesterol (25-HC), an oxysterol that represses cholesterol biosynthesis enzymes and promotes cholesterol transport outside the cells. The role of the oxysterol 25-HC on inflammation has also been extensively studied, linking alterations in lipid metabolism with neuroinflammation, one of the main pathological processes in ALS [35]. In fact, 25-HC has been previously suggested to be involved in the pathogenesis of ALS patients and mice [36]. The findings of several cholesterol derived oxysterols in the CSF and serum of ALS patients highlights a central role for altered oxysterol metabolism in ALS [28] and other motor neuron diseases, such as hereditary spastic paraplegia [37]. Interestingly, it seems that cholesterol biosynthesis is downregulated in several diseases related with motor dysfunction, as identified in an animal model of SCA2 and ALS, carrying an intermediate CAG expansion in the *ATXN2* gene [38]. More studies on the role of cholesterol pathways in motor neuron diseases are needed.

A recent review reported that there are at least two lipid metabolic pathways commonly altered in motor neuron diseases, especially in spastic paraplegia; the cholesterol/oxysterol and phosphatidylethanolamine biosynthesis pathways [39]. In this study, we have shown that the ceramides and the phosphatidylethanolamine biosynthesis pathways are also transcriptionally upregulated in the spinal cord tissue from SOD1 transgenic mice from early symptomatic disease stage. Our pathways analysis points towards an increase in the production of ceramides, mainly through the upregulation of the *HEXA*, *HEXB*, *GLB1*, and *GBA* genes of the ganglioside pathway, supporting the formation of ceramides as one of the earliest events in disease. This is consistent with the higher levels of ceramides found in the post mortem spinal cord tissue from ALS patients [17], in symptomatic disease stage mouse models of ALS, as well as in the spinal cord of mice at early disease stages [40]. In this regard, the inhibition of the glucosylceramide synthase has been used in *SOD1^G93A^* mice and resulted in an acceleration of the disease progression, and, on the contrary, the administration of the ganglioside GM3, product of the HEX enzyme activity, slowed disease progression [40]. The phosphatidylethanolamine biosynthesis pathways have also been found altered in ALS patients. In consequence, there has been some attempts to target the sphingosine pathways by administrating an analogue of sphingosine (fingolimod) using the same SOD1 mouse model, with a significant improvement in survival [41], and has been used in a phase II clinical trial (NCT01786174). Taken together, it seems that the upregulation of ceramide production through the upregulation of gangliosides, as well as the biosynthesis of phosphatidylethanolamine, are compensatory events, as the drugs used to enhance those already upregulated pathways showed some benefits.

Finally, eicosanoid pathways are found dysregulated from early stages of disease in our study. This group of lipids are essential for the production of precursors for the inflammatory response. The phospholipases are hydrolases that liberate the polyunsaturated fatty acid arachidonic acid (AA) from the glycerophospholipids of the membranes. In relation to ALS, the phospholipases A2 have been found upregulated in mice [42] and in ALS patients [43]. We have found an upregulation of one of the phospholipases A2 enzymes, the *PLA2G15* gene, which encodes a lysosomal phospholipase A2 enzyme that might be involved in the detoxification of reactivated phospholipids damaged by oxidative stress. This particular enzyme might act as a bridge between the ceramide pathway and the phospholipid pathway, as it has a transacylating function that can use short acyl ceramides as substrates. The arachidonic acid is the precursor of inflammatory process. Our study clearly points towards an upregulation of all the inflammatory pathways derived from this lipid from early disease stages, which is consistent with all the inflammatory markers that have been found in the spinal cord of ALS mice and patients from very early disease stages until the very end stages [44].

In summary, we conducted a comprehensive transcriptomic meta-analysis at an early and a late symptomatic disease stages in the spinal cord of SOD1 mice to determine the early transcriptional regulation of the well documented lipid alterations found in this mouse model and in ALS patients. The cholesterol, ceramides, and eicosanoid pathways are transcriptionally altered from early disease stages in the spinal cord of SOD1 mice. Further research is needed to determine the cellular contribution of the transcriptional alterations in the lipid regulation observed in the spinal cord. More efforts must be made into the understanding of the regulation of lipid metabolism in the CNS prior disease stages, and the relation among these lipid alterations, so we can find a major trigger for the observed lipids alterations, that can be therapeutically targeted.

## 4. Materials and Methods

### 4.1. Animals

The *SOD1^G93A^* mouse strain [B6.Cg-Tg(*SOD1*-G93A)1Gur/J)] carries a high copy number (approximately 25 copies) of the *SOD1* transgene, were obtained from Jackson Laboratories (Bar Harbor, Maine, USA) and were maintained on a C57BL/6J background (purchased from Charles River) in our lab. Mice were genotyped using conventional PCR, as well as controlling for the transgene copy number, using quantitative PCR (see list of primers in Appendix A). Mice were kept on auto ventilated cages with food ad libitum on a 12 h light–dark cycle. Mice were weighed weekly and humanely sacrificed before reaching the pre-established end point. Based on disease progression and the onset of weight loss by previous studies, the early symptomatic stage was selected at 90 days (P90). Similarly, we selected P60 as the pre-symptomatic disease stage and P120 as the late symptomatic disease stage. Female *SOD1^G93A^* mice and their control wild type (WT) littermates (*n* = 5 per group) were sacrificed at 90 days, and the lumbar region of the spinal cord was dissected and snap frozen for the RNA-sequencing analysis. Although the alterations are expected in both sexes, we used only females for the validation of gene expression experiments to be consistent with our RNA-seq data and to reduce variability. Still, for the meta-analyses, the datasets were coming from both sexes. Three different cohorts of female mice *SOD1^G93A^* and their wild-type littermates at 60, 90, and 120 days were used for gene expression validations (*n* = 3–5 per group). All animal procedures were approved by the ethical committee of animal care and use of the Hospital Clínico San Carlos and in accordance with the European and Spanish regulation (2010/63/EU and RD 1201/2005).

### 4.2. RNA Extraction and Sequencing

RNA was isolated from dissected lumbar region of spinal cords of wild-type and *SOD1^G93A^* female mice at 90 days, using Qiazol followed by the mini lipid tissue RNAeasy kit (Qiagen, Hilden, Germany). The RNA used for sequencing had a RIN value above 8–9 in the Bioanalyzer. The samples were sent to the company NIM Genetics (Madrid, Spain) for sequencing. The quality control of the samples was achieved with TapeStation (Agilent Technologies, Santa Clara, CA, USA) followed by quantification using the fluorometric system Qubit (Thermo Fisher Scientific, Waltham, MA, USA) cDNA libraries were made using TruSeq Stranded mRNA Library Prep and sequenced on NovaSeq 6000 (all Ilumina, Inc., San Diego, CA, USA) producing paired-end 100 bp reads.

### 4.3. RNA-seq Data Processing

Quality control of FastaQ files was performed using FastQC (https://www.bioinformatics.babraham.ac.uk/projects; accessed date November 2020). Low-quality reads (Phred quality score < 30) and reads too short (length < 30 pb) were removed using Fastp [45]. The alignment to the genome (mm10 mouse reference genome) was achieved using HISAT2 [46]. The expression quantification of genes was carried out using FeatureCounts [47]. Only uniquely mapped reads were used for the analysis of differential gene expression quantification with DESeq2 [48]. Raw p-values were adjusted by the Benjamini–Hochberg false discovery rate (FDR) method and the adjusted p-values less than 0.05 were considered statistically significant.

Volcano-plots, PCA analysis, and heatmaps were generated using R and the following packages: “DESeq2” [48] and “pheatmap” (https://CRAN.R-project.org/package=pheatmap; accessed date January 2021).

### 4.4. Functional Enrichment Analysis

A functional enrichment analysis (Table 1 and Table 2) was performed to identify up- and downregulated genes found in study 1 using the WEB-based Gene Set Analysis Toolkit (WebGestalt) [49]. We selected the Over-Representation Analysis (ORA) [50] method which performs a statistical evaluation of the fraction of genes in a particular pathway found among the set of genes.

The following parameters were selected. Organism: Mus musculus; Method: ORA; functional database: gene ontology + biological process: no redundant; gene list: type: gene name; upload: list of DEGs found in study 1; minimum number of genes for category: 5; multiple test adjustment: Benjamini–Hochberg; significant level: FDR (0.05).

### 4.5. RNA-seq Databases Selected for the Meta-Analysis

The databases with the mouse transcriptome information analysed in this study were downloaded from the NCBI GEO public data repository (http://www.ncbi.nlm.nih.gov/geo/; accessed date October 2020). The terms and/or their combinations used in the searching of databases were the following: “amyotrophic lateral sclerosis”, “RNA-seq”, “transcriptome”, “SOD1”, and “mouse”. The chosen databases had to meet all inclusion criteria: (1) the data had to be obtained from spinal cord of SOD1 mice and controls, (2) these transcriptome data had to be acquired by bulk RNA-sequencing, and (3) the ages had to be at early symptomatic (around 90 days) and late symptomatic (from 120 days). With this selective criteria, five more studies were added (studies 2 to 6) to our own study (study 1) to perform a meta-analysis. A summary of the datasets used is shown in Table 3.

### 4.6. Meta-Analysis of Combined RNA-seq Studies

Once FastQ files of each study (see Table 3) were downloaded, they were analysed following the same pipeline described in Section 4.3 RNA-seq Data Processing. DEGs obtained in each individual RNA-seq study were further combined in a meta-analysis to identify genes of differential expression across early symptomatic and late symptomatic studies. Both meta-analyses were performed through Fisher’s combined probability test [51] using R and the package “metaRNASeq” [22]. Raw p-values were adjusted by the Benjamini–Hochberg false discovery rate (FDR) method and the adjusted p-values less than 0.01 were considered as statistically significant.

### 4.7. Protein–Protein Interaction Network

A protein–protein interaction network was constructed using STRING [52] from the list of lipid DEGs found in the early symptomatic meta-analysis. The following parameters were selected: network type: full STRING network; required score: high confidence (0.7); FDR stringency: medium (5 percent).

### 4.8. Pathway Analysis

In order to analyse the most interesting lipid metabolism pathways found in this meta-analysis, we performed a heatmap showing the progressive increase or decrease in the Log2Foldchange of the genes related to these pathways among early symptomatic and late symptomatic studies. Heatmaps were performed using R and “pheatmap” package. The scheme representation of each route is a modified version from those designed on the Lipid Maps website (https://www.lipidmaps.org/resources/pathways/wikipathways.php; accessed date March 2021)

### 4.9. Quantitative PCR

RNA was isolated from dissected lumbar region of spinal cords of wild-type and *SOD1^G93A^* female mice at 60, 90, and 120 days (*n* = 3–5 per group), using Qiazol, followed by the mini lipid tissue RNAeasy kit (Qiagen, Hilden, Germany). Extracted RNA was treated with DNAse (Qiagen, Hilden, Germany).) to avoid carried over DNA contamination. cDNA was synthesized using the High Capacity cDNA RT kit (Thermo Fisher Scientific, Waltham, MA, USA) from 1 µg of RNA, including a negative control sample (without the addition of the enzyme). The real time reactions were run in triplicate with 25 ng of cDNA per 10 µL well, using Fast SyBr Green Master Mix on a 7900 Fast Machine (Thermo Fisher Scientific, Waltham, MA, USA). Primers were at a final concentration of 360 nM. Primers were designed to span exon–exon boundaries and validated prior to use, or extracted from already validated databases (PrimerBank). Fold changes were calculated using the 2^delta-delta- Ct method [53] using the 7500 Software v2.0.6 and normalized using S16 endogenous reference gene relative to a value of the WT group at P60 (for primer list, see Appendix A).

## Figures and Tables

**Figure 1 ijms-22-09553-f001:**
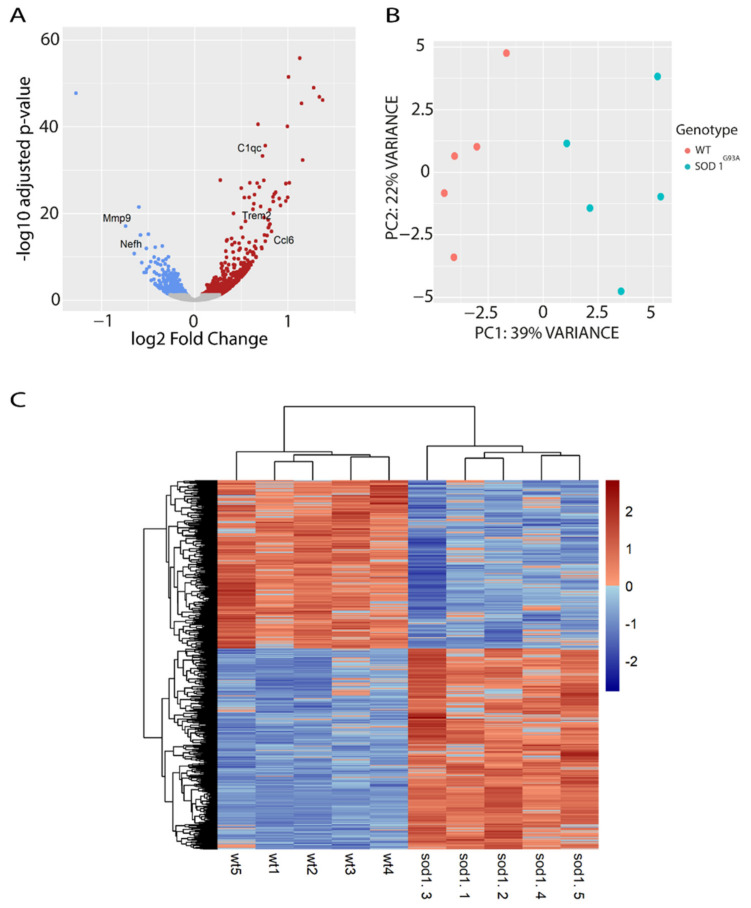
Transcriptional analysis of the spinal cord of *SOD1^G93A^* females and WT littermates at 90 days of age (P90). (**A**) Volcano-plot showing the fold changes (up and downregulated) of the DEGs found in the analysis. Red dots indicate significantly upregulated genes, blue dots indicate significantly downregulated genes, and grey dots indicate genes with no significant change. (**B**) PCA plot showing the percentage (%) of variability between samples due to the genotype effect. (**C**) The heatmap graph shows a consistent cluster of DEGs up- and down-regulated between the genotypes. Each row represents a gene, and each column a sample. The expression level of each gene was normalized using row z-score. The genes in red indicate higher expression level and blue indicates lower expression level.

**Figure 2 ijms-22-09553-f002:**
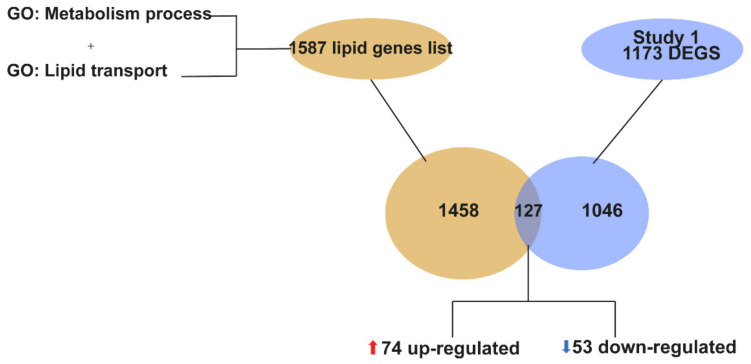
Graphical representation of the identification of genes related to lipid metabolism among the total of DEGs identified at P90. The lipid gene list was obtained by joining genes annotated in the GO terms “lipid metabolism process “(GO:0006629) and “lipid transport” (GO:0006869), resulting in a total of 1587 lipid genes annotated. The Venn diagram is showing that out of the 1173 DEGs identified in our study, a total of 127 DEGs are lipid genes, of which 74 are upregulated and 53 downregulated.

**Figure 3 ijms-22-09553-f003:**
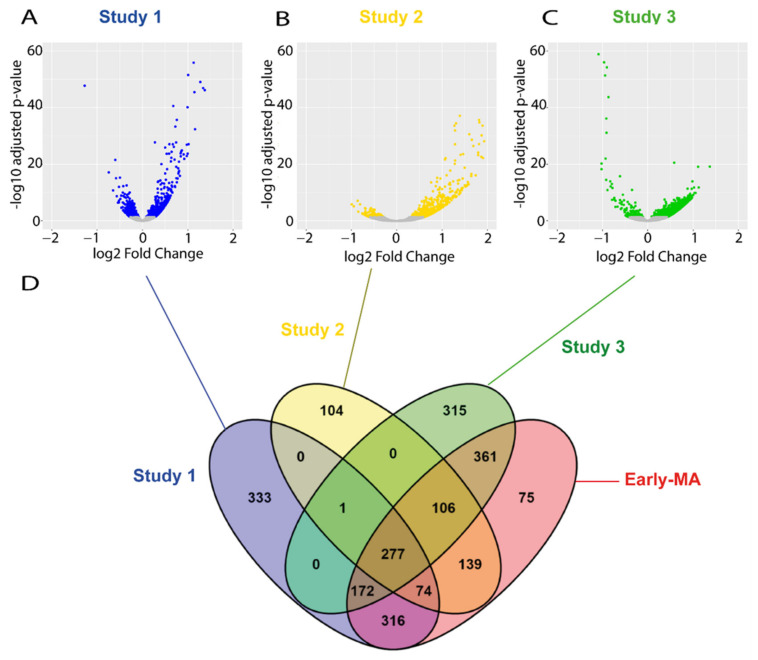
Meta-analysis of early symptomatic RNA-seq studies. (**A**–**C**) Volcano-plots showing the dysregulated genes found on each of the three RNA-seq studies selected at P90. Blue, yellow, and green dots indicate DEGs found in the study 1, 2, and 3, respectively. Grey dots indicate genes with no significant change. (**D**) A Venn diagram illustrating the overlapping DEGs across the three studies and the result of the meta-analysis (Early MA). The meta-analysis (Early MA) was performed using Fishers combined probability test where raw p-values were adjusted by the Benjamini–Hochberg false discovery rate (FDR) method and the adjusted p-values less than 0.01 were considered as statistically significant.

**Figure 4 ijms-22-09553-f004:**
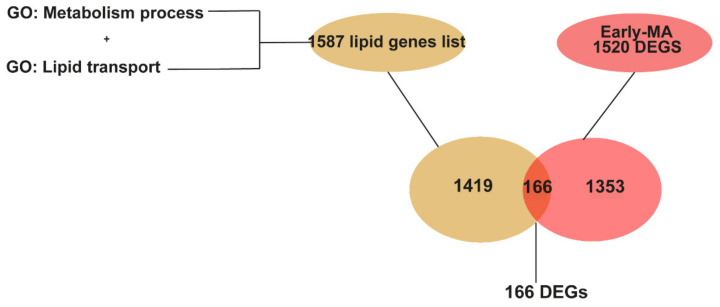
Graphical representation of the identification of genes related to lipid metabolism among the DEGs in the Early MA. The list of genes related to lipid metabolism was obtained from genes annotated in the GO term “lipid metabolism process” (GO:0006629) and “lipid transport” (GO:0006869). A Venn diagram showing that 166 DEGs from the Early MA are lipid genes.

**Figure 5 ijms-22-09553-f005:**
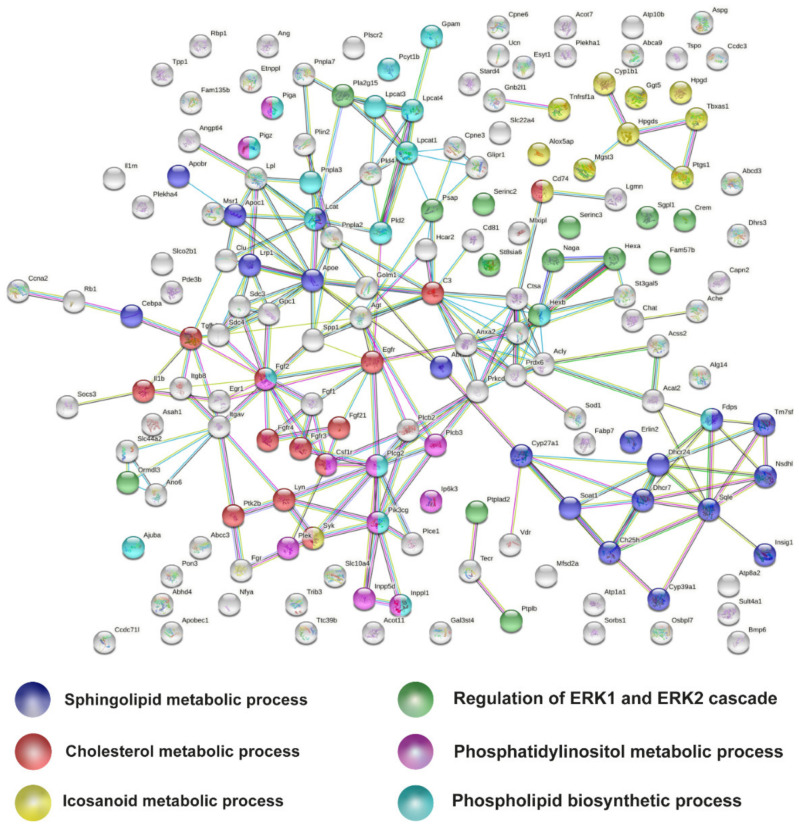
Protein-protein interaction network of lipid DEGs from early symptomatic meta-analysis. The six relevant lipid GO terms are colour-coded as follows: sphingolipid metabolic processes in blue, cholesterol metabolic processes in red, eicosanoid metabolic processes in yellow, phospholipid biosynthetic processes in cyan, regulation of ERK1 and ERK2 cascade in green, and phosphatidylinositol in pink.

**Figure 6 ijms-22-09553-f006:**
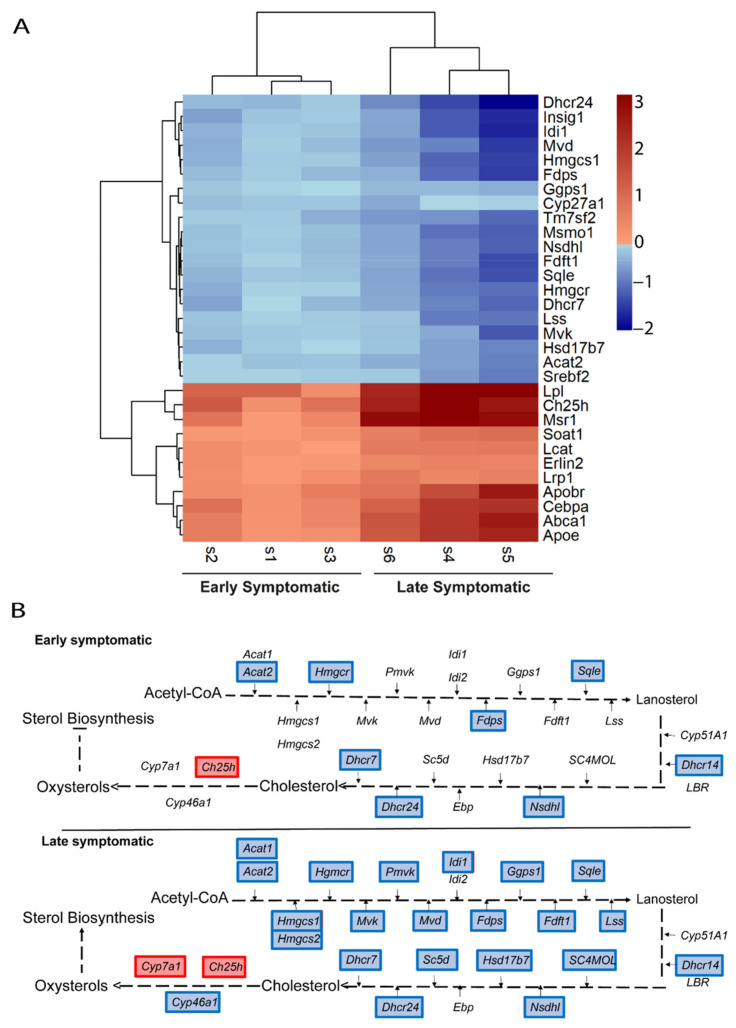
Comparative transcriptional alterations in cholesterol metabolic pathways in the spinal cord of SOD1 mice at early and late symptomatic disease stages. (**A**) A heatmap of the combination of the six studies (three early symptomatic and three late symptomatic) showing the expression level of the altered genes that participate in cholesterol metabolic pathways. The heatmap colours indicate the expression levels by the log2-fold-change. Each column represents an individual RNA-seq study, and each row represents one gene. (**B**) The cholesterol biosynthesis and transport pathways are represented, comparing the results from the Early MA and the Late MA. Coloured genes indicate that the gene was significantly altered in the meta-analysis. Blue indicates downregulation and red indicates upregulation.

**Figure 7 ijms-22-09553-f007:**
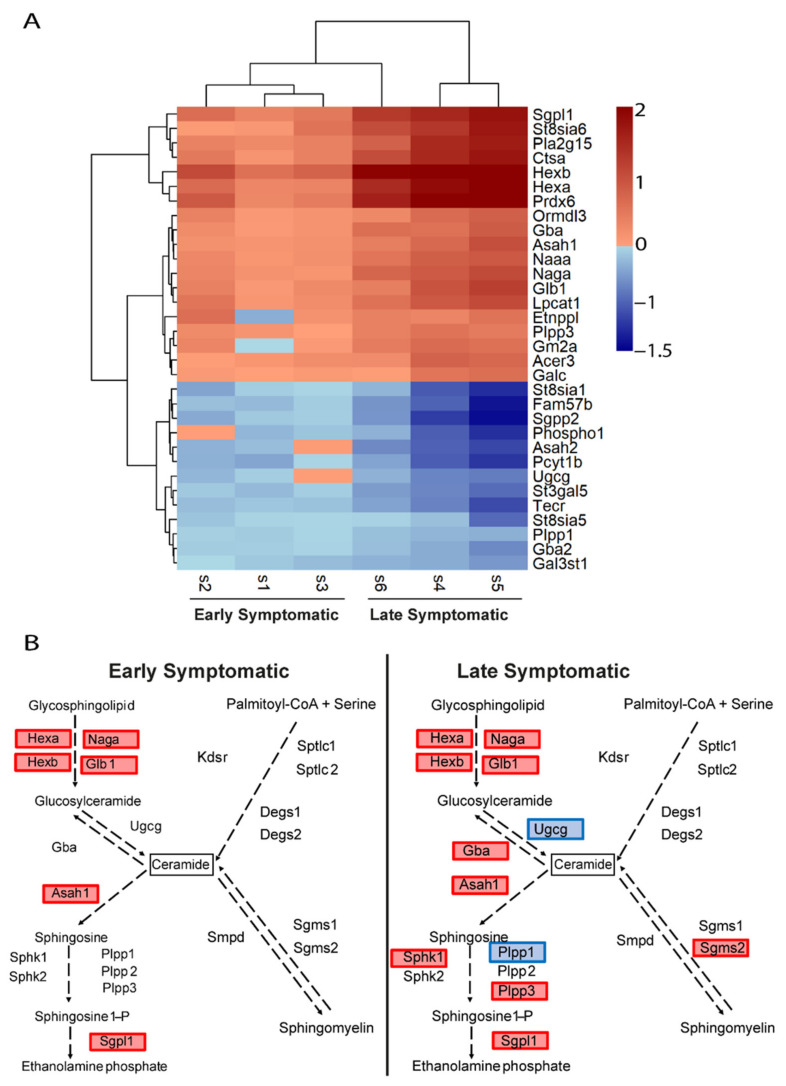
Comparative transcriptional alterations in glycosphingolipids and ceramides metabolic pathways in the spinal cord of SOD1 mice at early and late symptomatic disease stages. (**A**) The heatmap of the combination of the six transcriptomic studies (three early symptomatic and three late symptomatic) shows the expression level of the altered genes that participate in the glycosphingolipids metabolic pathways. The heatmap colours indicate the expression levels by the log2-fold-change. Each column represents an individual RNA-seq study, and each row represents one gene. (**B**) The sphingolipids and ceramides biosynthesis pathways are represented, comparing the results from the Early MA and the Late MA. Coloured genes indicate that the gene was significantly altered in the meta-analysis. Blue indicates downregulation and red indicates upregulation.

**Figure 8 ijms-22-09553-f008:**
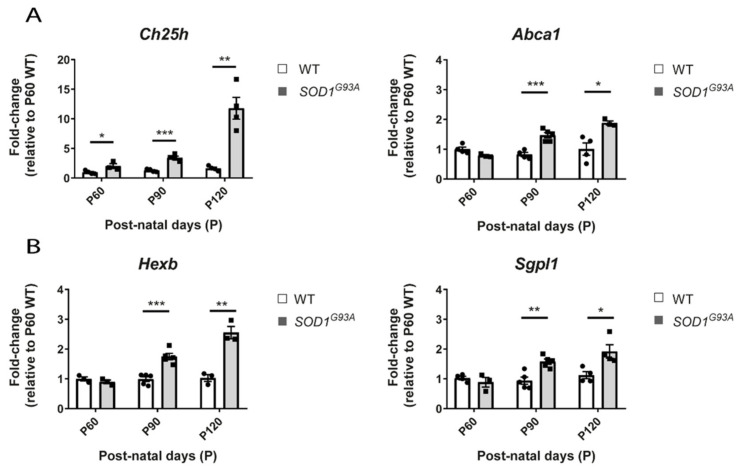
Gene expression level of selected lipid genes by qPCR in the lumbar region of the spinal cord of WT and *SOD1^G93A^* females at different disease stages. Graphs showing the gene expression quantification of several genes by qPCR, in the spinal cord of WT and *SOD1^G93A^* females at 60 days (WT *n* = 4, *SOD1^G93A^* = 3), at 90 days (WT *n* = 5, *SOD1^G93A^* = 5), and 120 days of age (WT *n* = 4, *SOD1^G93A^* = 4). The values are relative to the level of one WT from the P60 group. (**A**). Graphs showing the gene expression quantification of two genes from the cholesterol metabolism pathways, the *Ch25h* and *Abca1*. The data shows the progressive upregulation of *Ch25h* from P60. (**B**) Graphs showing the gene expression quantification of two genes, *Hexb* and *Sgpl1,* from the sphingolipid pathway. The significant upregulation in these genes is observed from P90. Data are represented as the mean +/− standard error of the mean. *p*-value * < 0.05, ** < 0.01, and *** < 0.001.

**Table 1 ijms-22-09553-t001:** Enriched biological processes (Gene Ontology-GO) of DEGs found.

	**Upregulated Genes**	
**GO Term**	**Biological Process**	**FDR-Value ^1^**
GO:0002237	response to molecule of bacterial origin	<1 × 10^−13^
GO:0071216	cellular response to biotic stimulus	<1 × 10^−13^
GO:0002274	myeloid leukocyte activation	<1 × 10^−13^
GO:0002443	leukocyte mediated immunity	1.80 × 10^−13^
GO:0001819	positive regulation of cytokine production	1.06 × 10^−12^
GO:0006909	phagocytosis	1.37 × 10^−12^
GO:0070661	leukocyte proliferation	5.78 × 10^−12^
GO:0002250	adaptive immune response	2.01 × 10^−11^
	**Downregulated Genes**	
**GO Term**	**Biological Process**	**FDR-Value ^1^**
GO:0099177	regulation of trans-synaptic signaling	2.58 × 10^−6^
GO:0042391	regulation of membrane potential	4.27 × 10^−5^
GO:0051656	establishment of organelle localization	1.29 × 10^−4^
GO:0061564	axon development	1.29 × 10^−4^
GO:0051648	vesicle localization	1.57 × 10^−4^
GO:0099504	synaptic vesicle cycle	1.64 × 10^−4^
GO:0006836	neurotransmitter transport	3.4 × 10^−4^
GO:0006887	exocytosis	3.99 × 10^−4^

^1^ FDR-adjusted *p*-value.

**Table 2 ijms-22-09553-t002:** Enriched biological processes related to lipid (Gene Ontology-GO) of DEGs found.

GO Term	Biological Process	FDR-Value ^1^
GO:0071402	cellular response to lipoprotein particle stimulus	2.86 × 10^−6^
GO:0055094	response to lipoprotein particle	1.11 × 10^−5^
GO:0036314	response to sterol	5.13 × 10^−4^
GO:0019216	regulation of lipid metabolic process	0.009
GO:0006638	neutral lipid metabolic process	0.037
GO:0046486	glycerolipid metabolic process	0.039

^1^ FDR-adjusted *p*-value.

**Table 3 ijms-22-09553-t003:** Summary of GEO datasets used for the meta-analyses.

Study	Dataset	Platform	Tissue	Age	Sample Size
Study 1	**Our study**	Ilumina NovaSeq 6000	Mouse spinal cord (lumbar)	90 days	5 WT 5 *SOD1^G93A^* (females)
Study 2	GSE43879	Ilumina Genome Analyzer II	Mouse spinal cord (unspecified region)	95 days	2 WT 2 *SOD1^G93A^* (both sexes)
Study 3	GSE106364	Ilumina HiSeq 4000	Mouse spinal cord (lumbar)	90 days	5 WT 5 *SOD1^G86R^* (females)
Study 4	GSE106803	Ilumina HiSeq 2000	Mouse spinal cord (unspecified region)	5 months	3 WT 3 *SOD1^G93A^* (both sexes)
Study 5	GSE100888	Ilumina HiSeq 2000	Mouse spinal cord (lumbar)	5 months	4 WT 4 *SOD1^G93A^* (both sexes)
Study 6	GSE433879	Ilumina Genome Analyzer II	Mouse spinal cord (unspecified region)	4 months	2 WT 2 *SOD1^G93A^* (both sexes)

## Data Availability

RNA-seq data as well as the meta-analyses are in the process of being uploaded at the Gene Expression Omnibus (GEO) database repository.

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
