# Peer review of "A Transcriptomic Meta-Analysis Shows Lipid Metabolism Dysregulation as an Early Pathological Mechanism in the Spinal Cord of SOD1 Mice"

_ijms, 2021, doi:10.3390/ijms22179553_

Round 1

Reviewer 1 Report

attached

Reviewer 2 Report

The proposed study by Fernandez-Beltran et al. is well designed and presented. It is an elegant work coupling original RNAseq data with the public databases meta-analysis, providing useful information about the ASL animal model, the ASL pathology and the use of transcriptomic techniques.

The background, methods and all the study workflow are clear and well described.

My only suggestion is to add a comment about the use of the female animals in the experiments:

  • Why only females have been used?
  • Did the authors check for the estrus cycle before the sacrifice?
  • Is there any possible bias linked to the interaction between hormones fluctuations and the targets of the study or the disease? For example, is the expression of the genes involved in the lipid dynamics or the lipid dynamics themselves influenced by the estrus cycle?
  • Is there any description of gender-related differences at molecular level or pathology progression in human patients?

Reviewer 3 Report

In this study, RNA sequencing was performed on the spinal cords of pre-symptomatic female SOD1G93A mice and their wild type littermates in order to identify changes in gene expression occurring at an early disease stage. In particular, the authors were interested in identifying changes occurring in lipids. The results of this analysis were then combined with publicly available RNA sequencing datasets to perform a more robust transcriptomic meta-analysis. Several lipid metabolic pathways were shown to be dysregulated in the pre-symptomatic mice worsening in the symptomatic mice, and select differentially expressed genes were validated using qPCR.

I really enjoyed reading this manuscript. The authors did an excellent job at explaining each step of their workflow including clear rationale for their decisions. The results are very interesting and provide clear evidence of a role for dysregulated lipid metabolism in the early stages of ALS.

Some minor corrections are required as outlined below.

  1. Overall, the manuscript should be checked for English language and style. Several spelling and formatting mistakes were identified including but not limited to:
    • Section 4.1, 6th line – autoventilated not autoventiladed
    • Section 2.2, last line on page 5 – pre-symptomatic not pre-synmptomatic
    • Section 2.6, 2nd line – progressively not progressively
    • Discussion – biosynthesis not biosinthesis
    • Table S2 heading – the not ‘de’
    • Supplementary figure 2 – check for consistency between graphs e.g., 2.0 vs 2 etc
    • Supplementary figure 3 – lowercase sod1 used
    • The abbreviated term DEGS was used prior to the full term which occurs first on page 4
  1. Study 3 from the pre-symptomatic meta-analysis used mice with a SOD1G86R . Can the authors comment on how this mutation compares to the SOD1G93A mutation, and the appropriateness for including it together with the other studies that assessed SOD1G93A mice.
  2. Supplementary Fig 2
    • Should the mean WT value be 1.0? It is less than 1.0 for Insig1.
    • Gene names in italics?
    • It would be useful to include on the figure and/or legend that this data is generated from pre-symptomatic mice.
  1. Tables and figures
    • Some tables, in particular table 1 are a bit blurry and look like they have been scanned into the manuscript on low quality.
    • Most figures have aspects that are blurry. In particular, the values on the axes of figures 1 and 3, and the text in figures 5, 6 and 7 is very difficult to read.

Round 2

Reviewer 1 Report

The authors responded to most of my comments. They included new data, clarified procedures and made significant other changes to their presentation.